# Localization-aware Deep Learning Framework for Statistical Shape Modeling Directly from Images

**Janmesh Ukey**      JANMESH@SCI.UTAH.EDU**, Shireen Elhabian**      SHIREEN@SCI.UTAH.EDU
*Scientific Computing and Imaging Institute, University of Utah, Salt Lake City, Utah-84112, USA ,*
*School of Computing, University of Utah, Salt Lake City, Utah-84112, USA*

**Editors:** Accepted for publication at MIDL 2023

## Abstract

Statistical Shape Modelling (SSM) is an effective tool for quantitatively analyzing anatomical populations. SSM has benefitted largely from advances in deep learning where statistical representations of anatomies (e.g., point distribution models or PDMs) are inferred directly from images, alleviating the need for a time-consuming and expensive workflow of anatomy segmentation, shape registration, and model optimization. Nonetheless, to date, existing deep learning methods do not consider the rigid pose transformation of shapes or anatomy of interest. They also require a tight bounding box to be defined over the image of anatomy-of-interest before feeding the image to the deep network for network training and inference. In this paper, we propose a deep learning framework that simultaneously detects and segments the anatomy of interest, estimate the rigid transformation with respect to the population mean (average) using a spatial transformer, and estimates the corresponding statistical representation of that anatomy, all directly from unsegmented 3D image without the need for any additional supervision. Furthermore, we leverage the segmentation task to provide an attention model for the sub-network that estimates shape representation, giving more accurate shape statistics for shape analysis.

**Keywords:** Statistical Shape Modeling, Deep Learning, Anatomy Segmentation

## 1. Introduction

Statistical Shape Modeling (SSM) effectively quantifies and characterizes the variations of anatomical shapes. SSM has enabled various biomedical and clinical applications in neuroscience (Gerig et al., 2001), orthopedics (Harris et al., 2013; Atkins et al., 2019), cardiology and biological phenotyping; such as implant design (Kozic et al., 2010) and growth modeling (Datar et al., 2009; Atkins et al., 2017). A popular approach for shape analysis is using landmarks defined by correspondences across the population of shapes. Most recent works (Sarkalkan et al., 2014; Cates et al., 2017; Tóthová et al., 2018) use dense correspondences placed automatically i.e., point distribution models (PDMs) via optimization for shape modeling that can then be turned into low-dimension shape representation by principal component analysis (PCA), thus reducing the need for expert intervention to mark distinct features, which can often be time and cost-consuming.

Conventional SSM methods still require an expert-driven workflow of anatomy segmentation, shape registration, optimization of population-level shape representations, and significant parameter tuning. This has motivated the recent surge of deep learning based methods that estimate statistical representations of shape directly from images (Bhalodia et al., 2018b,a; Adams et al., 2020; Adams and Elhabian, 2022; Tao et al., 2022; Huang

et al., 2017; Milletari et al., 2017; Xie et al., 2016; Zheng et al., 2015; Raju et al., 2022).

Such deep networks learn to regress shape information directly from images while learning complex functional mappings and incorporating prior shape knowledge. These methods remove most of the manual overhead needed to estimate an SSM from unsegmented images once a deep network is trained and have been shown to perform statistically similarly to traditional SSM methods in downstream tasks (Bhalodia et al., 2018b). Nonetheless, these methods still require input image pre-processing in the form of indicating the anatomy location, defining a tight bounding box around the anatomy of interest, and aligning the given image to the model's mean. Specifically, images are assumed to be cropped around the anatomy of interest (i.e., manual delineation of the bounding box of the anatomy) in training and during model inference. These methods also do not provide rigid pose information; thus, they are only applicable in constrained and limited setups. This is a barrier to deploying these methods as a fully automated alternative to conventional SSM methods.

In this paper, we propose LocalizedSSM, an end-to-end deep learning framework that simultaneously extracts the tight bounding box over the shape of interest from a complete image, transforms it to match the mean shape, and uses this to extract statistical shape representation. LocalizedSSM comprises of three subnetworks that are trained end-to-end; a segmentation subnetwork, a registration subnetwork, and a shape regression subnetwork. Specifically, LocalizedSSM first performs anatomy segmentation using the segmentation subnetwork to localize the anatomy and provide an attention map for the shape regression subnetwork to estimate shape statistics from images. Then a spatial transformer does an affine registration of the cropped region (based on the output of the segmentation subnetwork) to match the mean shape, which is then passed to the shape regression subnetwork that learns the functional mapping from image features to a low-dimensional shape representation. Given a new image, LocalizedSSM alleviates any manual input required for model inference. In addition, as LocalizedSSM is a multi-task network, it provides significant accuracy improvements compared to using cropped images for estimating shape descriptors. Moreover, LocalizedSSM is also orthogonal to existing deep learning methods (Adams et al., 2020; Adams and Elhabian, 2022; Tao et al., 2022; Huang et al., 2017; Xie et al., 2016; Zheng et al., 2015), any of which can be integrated as the shape regression subnetwork in a straightforward manner.

## 2. Methods

Given a dataset of $N$ samples, we denote the unsegmented 3D images $\{\mathbf{I}_n\}_{n=1}^N$ where $\mathbf{I}_n \in \mathbb{R}^{H \times W \times D}$. The corresponding segmentations are denoted $\{\mathbf{S}_n\}_{n=1}^N$ where $\mathbf{S}_n \in \mathbb{R}^{H \times W \times D}$. The set of corresponding PDMs comprised of $M$ world 3D correspondence points is denoted $\{\mathbf{p}_n\}_{n=1}^N$ where $\mathbf{p}_n \in \mathbb{R}^{3M}$, which are essentially coordinates representing the population-level statistical shape information after removing global alignment differences across samples in the given cohort (check 3 for how the correspondences were generated). The proposed network maps the volumetric image $\mathbf{I}_n$ to correspondences $\mathbf{p}_n$ in an end-to-end manner.

First $\hat{\mathbf{S}}_n$ is predicted from $\mathbf{I}_n$ via a segmentation sub-network and the excess background is cropped out, resulting in $\hat{\mathbf{S}}'_n$. Next, a spatial transformer sub-network does an affine registration of the cropped $\hat{\mathbf{S}}'_n$ to match the mean shape. The output of this step is denoted $\hat{\mathbf{S}}''_n$. The cropped, aligned segmentation, $\hat{\mathbf{S}}''_n$, is then passed to a shape regression sub-network

to output predicted correspondences $\hat{\mathbf{p}}_n$. An overview diagram is provided in Figure 1 and the sub-networks are described in the following sections.

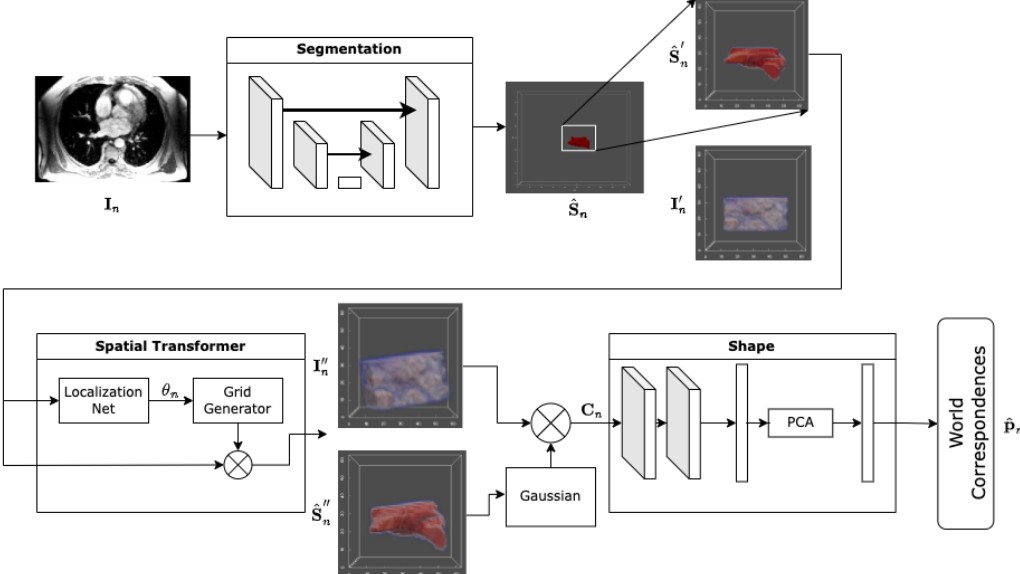

**Figure 1:** *LocalizedSSM Network Architecture.*

### 2.1. Segmentation sub-network

The segmentation sub-network utilizes a 3D UNet (Ronneberger et al., 2015; Çiçek et al., 2016) architecture to segment the anatomy of interest from the volumetric image (ı.e., predict $\mathbf{S}_n$ from $\mathbf{I}_n$). A binary cross entropy loss ($L_{seg}$) in used for supervising the segmentation sub-network. Architecture details can be found in the supplementary materials.

$$L_{seg} = -\frac{1}{N}\sum_{i=1}^{N}\mathbf{S}_n \cdot \log\hat{\mathbf{S}}_n + (1 - \mathbf{S}_n) \cdot \log(1 - \hat{\mathbf{S}}_n); \tag{1}$$

The excess background is then cropped from the segmentation, resulting in $\hat{\mathbf{S}}'_n$. This identical cropping is also applied to the input image, resulting in $\mathbf{I}'_n$.

### 2.2. Registration sub-network

The registration sub-network employs a spatial transformer (Jaderberg et al., 2015) to enhance the model's geometric invariance by transforming the segmented shape. Shape is defined by the characteristic that remains after removing all global geometrical information from an object. To this end, we select a representative shape to which every shape is matched via a rigid transformation. The registration sub-network learns a rigid transformation (parameterized by $\theta_n$) from $\hat{\mathbf{S}}'_n$ to the representative shape, $\overline{\mathbf{S}}$. In our experiments, the medoid shape is used as the reference $\overline{\mathbf{S}}$; where the mediod is selected by first computing

the average segmentation and then selecting the training shape nearest to it. $\overline{\mathbf{S}}$ is used as a target for the supervised spatial transformer to make the shape regression network focus on estimating shape features that are invariant to Euclidean transformations.

The learned rigid spatial transform, $\theta_n$, is then applied to the cropped segmentation $\hat{\mathbf{S}}'_n$, resulting in $\hat{\mathbf{S}}''_n$. This transform is additionally applied to the cropped input image $\mathbf{I}'_n$, resulting in $\mathbf{I}''_n$. The registration sub-network optimized MSE loss ($L_{reg}$) between mean shape, $\overline{\mathbf{S}}$ and transformed cropped anatomy segmentation $\hat{\mathbf{S}}''_n$.

$$L_{reg} = \frac{1}{N}\left(\overline{\mathbf{S}} - \hat{\mathbf{S}}''_n\right)^2;  \tag{2}$$

### 2.3. Attention Map

We apply a Gaussian Filter $\mathbf{G}$ (kernel size $= 5$ , $\sigma = 2$) on the transformed cropped anatomy segmentation ($\hat{\mathbf{S}}''_n$), which acts as an attention map and is added with transformed cropped anatomy image ($\mathbf{I}''_n$) resulting in $\mathbf{C}_n$.

$$\mathbf{C}_n = (\mathbf{G} \circledast \mathbf{I}_n) \cdot \max(\mathbf{I}''_n) + \mathbf{I}''_n;  \tag{3}$$

This puts more emphasis on the boundary of shape making the work of shape sub-network network easier.

### 2.4. Shape sub-network

We based our shape sub-network on DeepSSM (Bhalodia et al., 2018a). It is a set of 5 convolution layers and 2 fully-connected layers that takes the result from the Gaussian Filter ($\mathbf{C}_n$) and regress to the PCA scores $\mathbf{z}_n$. The PCA score is passed through another linear fully-connected layer with the weights fixed as the eigenvectors and the bias as the mean shape (precomputed on training set world correspondences and hence provides and implicit shape prior), giving us an estimate of the world correspondences $\mathbf{p}_n$. The shape sub-network optimizes the MSE loss ($L_w$) between the estimated and the original world correspondences.

$$L_w = \frac{1}{3M}(\hat{\mathbf{p}}_n - \mathbf{p}_n)^2;  \tag{4}$$

The final loss is given by -

$$L = \lambda_{seg}L_{seg} + \lambda_{reg}L_{reg} + \lambda_w L_w;  \tag{5}$$

The proposed network is trained in a stepwise manner, initially only training for segmentation ($\lambda_{seg} = 100$, $\lambda_{reg} = \lambda_w = 0$) for the first ten epochs. Then the weight corresponding to the registration loss ($\lambda_{reg}$) is increased, and the network is trained for ten more epochs. Finally, incrementally increasing $\lambda_w$ until a certain value. Overall we train the network for 200 epochs; PyTorch is used in constructing and training the network with Adam (Kingma and Ba, 2014) optimizer and a learning rate of 2e-3 for the first ten epochs, then 1e-5 with step learning rate decay. The weights of the network are initialized by He initialization (He et al., 2015).

## 3. Results

We compare our results with two baseline approaches (Bhalodia et al., 2018a, 2021), DeepSSM$_c$ and DeepSSM$_f$. In DeepSSM$_f$ we use full images (as in our work) for training, whereas in DeepSSM$_c$, we crop the image to extract the anatomy as a pre-processing. By comparing these baselines, we showcase that LocalizedSSM performs better or comparatively to the networks trained separately. For each dataset, we evaluate the performance of each model by comparing the predicted world points to their ground truths and computing the root mean squared error (RMSE) averaged over each dimension. We also reconstruct mesh from the predicted and ground truth points and evaluate the surface-to-surface distance, which is essentially the average Haussdorff distance.

### 3.1. Supershapes

We used a 3D supershapes dataset as a proof of concept. Supershapes can be parametrized by three variables, two define the curvature, and the last defines the number of lobes. We generate 2000 shapes with random curvature and number of lobes between 4-10 chosen randomly. These shapes are randomly oriented and positioned in a 128x128x128 size image. Additive Gaussian noise is added to the images, and the images are blurred with a Gaussian filter to mimic diffuse shape boundaries. We form the initial PDM with 32 points from complete data using the Shapeworks software (Cates et al., 2017) and use 6 PCA components to capture 72% of shape variability. We use 1800 examples for training and the remaining for testing.

The box plots for unseen test data representing the Euclidean distance error per-point

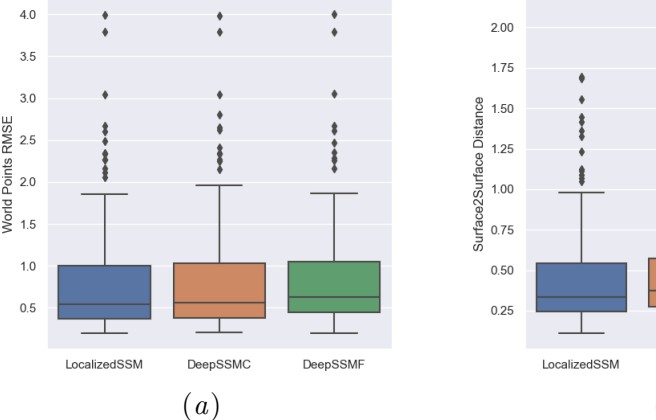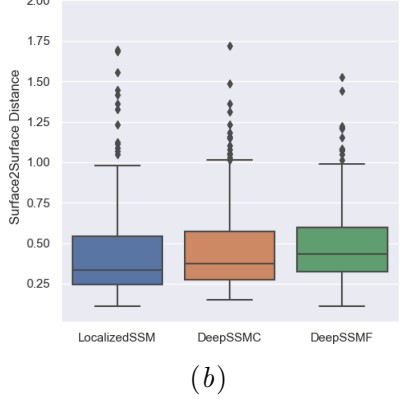

$(a)$ $(b)$

**Figure 2:** Supershapes. **(a) RMSE for world points (mm)**: *Euclidean distance between ground truth points and points obtained from predictions across all test samples.* **(b) Surface-to-surface distance (mm)**: *Computed between reconstructed mesh using ground truth points and points obtained from predictions across all test samples.*

per-shape is shown in Figure 2($a$); and the one representing surface-to-surface distance for reconstructed mesh is shown in Figure 2($b$). We can see that our method performs

comparably or better to the baselines. We have also provided the surface-to-surface distance visualization of reconstructed mesh for best, median and worst cases in the test set in Figure 3.

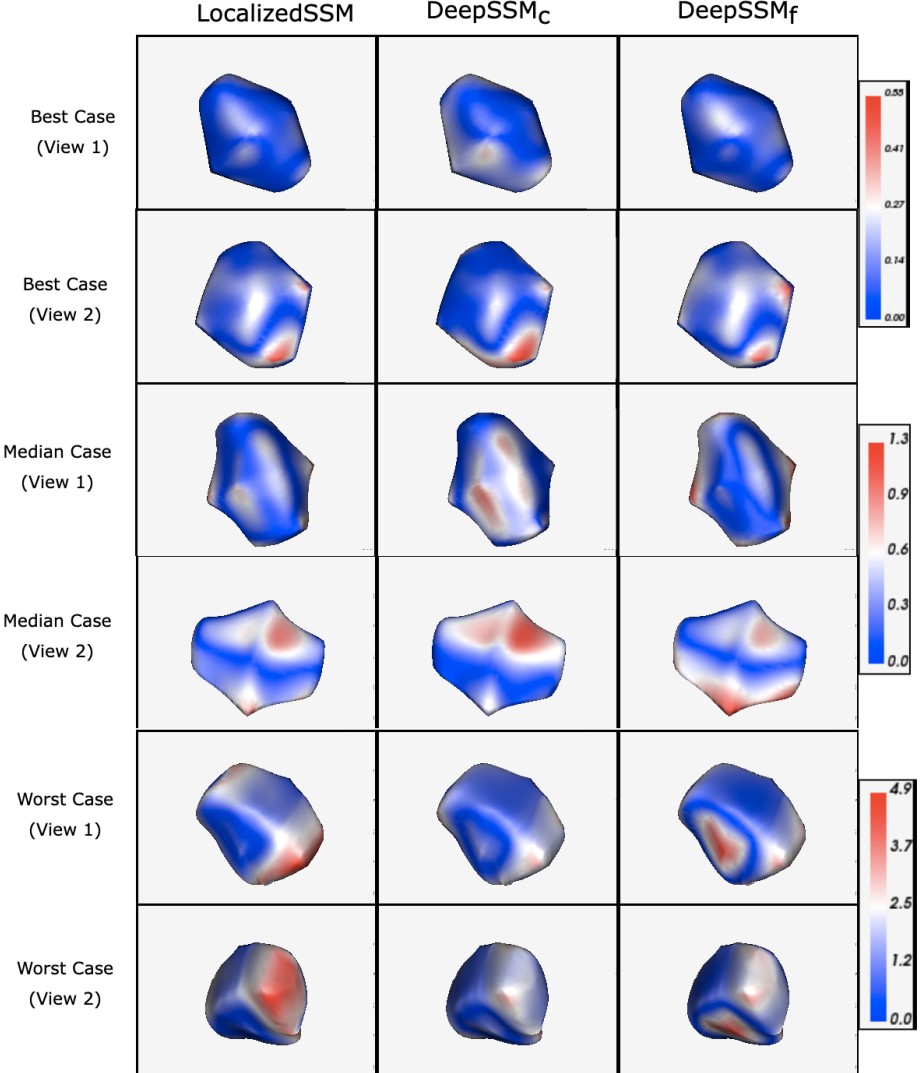

**Figure 3:** Supershapes. **Surface to surface distance (mm) visualization**: *Shape reconstruction error interpolated as a heatmap on ground truth reconstructed meshes. Best, Median and Worst case result on LocalizedSSM.*

## 3.2. Left Atrium

We apply our network on the Left Atrium MRI dataset. The data has a total of 1006 images that vary significantly in intensity and quality, more so due to topological variants pertaining to pulmonary veins arrangements. We divide the data into train and test sets. We form the initial PDM with 1024 points using the Shapeworks software (Cates et al.,

2017) from training data (914 examples) and use 300 PCA components to capture 97% of shape variability. Note that the test data is kept separate while generating the initial PDM; this is done to ensure that the test data remains completely unseen.

The box plots for unseen test data representing the Euclidean distance error per-point

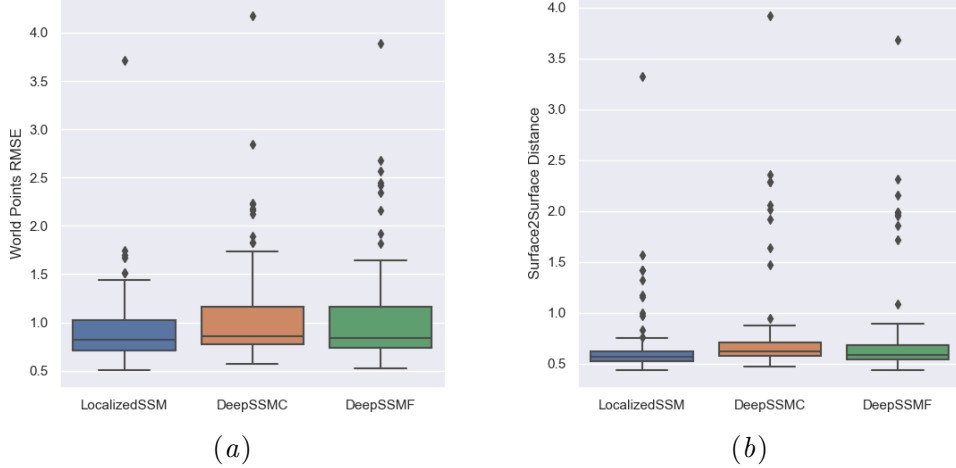

$(a)$ $(b)$

**Figure 4:** Left Atrium. **(a) RMSE for world points (mm)**: *Euclidean distance between ground truth points and points obtained from predictions across all test samples.* **(b) Surface-to-surface distance (mm)**: *Computed between reconstructed mesh using ground truth points and points obtained from predictions across all test samples.*

per-shape is shown in Figure 4($a$); and the one representing surface-to-surface distance for reconstructed mesh is shown in Figure 4($b$). We can see that our method performs comparably or better to the baselines, primarily as localizing the anatomy provides a strong prior for identifying world points. We have also provided the surface-to-surface distance visualization of reconstructed mesh for best, median and worst cases in the test set in Figure 6.

We also compare Left Atrium to another baseline where cropped and aligned images are used to train DeepSSM (DeepSSM_CA); LocalizedSSM performs comparably or even better than the lower bound achieved with cropped and aligned images (DeepSSM_CA).

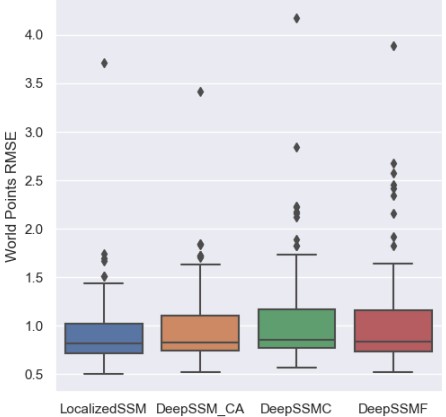

**Figure 5:** RMSE for world points (mm)

| Index | Value |
|-------|-------|
| Dice Score for Segmentation | 0.771 |
| Normalized Cross Correlation Loss for Registration | 0.0278 |

**Table 1:** Segmentation and Registration Performance for the Left Atrium Dataset

## 4. Conclusion

LocalizedSSM is an end-to-end deep learning framework for extracting the population shape representation in the form of world correspondence. It offers a novel approach for deep learning based shape modeling that simultaneously detects and segments the anatomy of interest, performs rigid transformation with respect to mean shape, and predicts the statistical representation of the anatomy. Our work alleviates any manual input required for model inference while also significantly improving the accuracy. We compare our work to the baseline DeepSSM models and find that the qualitative and quantitative performance of the proposed model is better or comparable with the baselines on two different datasets. Although the proposed work has some limitations, including that it does not provide local correspondences (i.e., in the coordinate frame of the original image), which might be useful in several downstream tasks Additionally, the proposed approach is a supervised method, and it requires software such as ShapeWorks to form the initial PDM for training and hence is dependent on the accuracy of the reference method (Goparaju et al., 2022) used. To the best of our knowledge, this paper is the first of its kind to introduce localization awareness for statistical shape modeling. Our results demonstrate that localization and registration are essential for effective shape modeling. Additionally, the proposed method can be readily integrated with other deep learning based shape modeling methods by replacing the shape regression sub-network and could improve upon the accuracy of those methods.

## Acknowledgments

This work was supported by the National Institutes of Health under grant numbers NIBIB-U24EB029011, NIAMS-R01AR076120, NHLBI-R01HL135568, NIBIB-R01EB016701. The content is solely the responsibility of the authors and does not necessarily represent the official views of the National Institutes of Health. The authors would like to thank the University of Utah Division of Cardiovascular Medicine for providing left atrium MRI scans and segmentations from the Atrial Fibrillation projects.

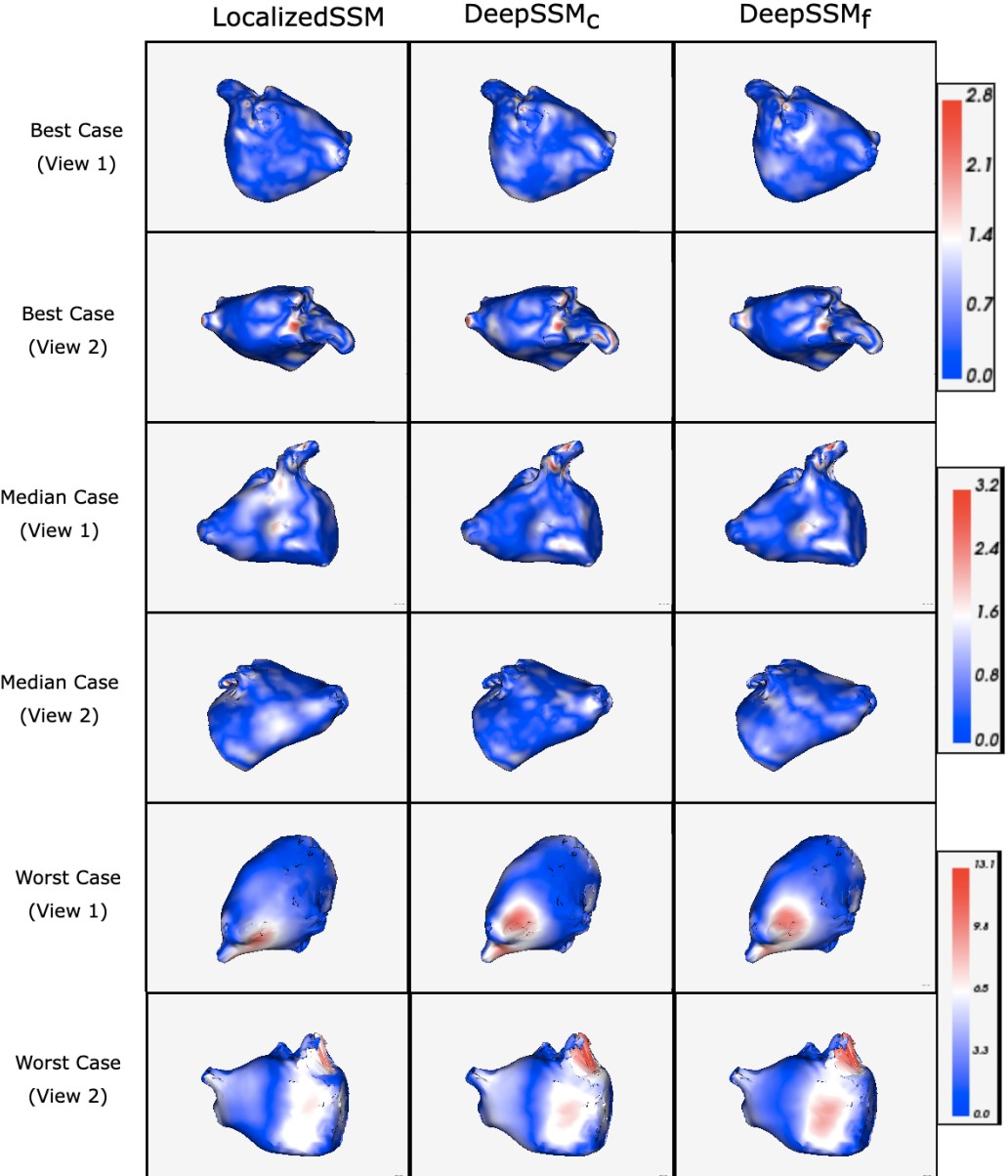

**Figure 6:** Left Atrium. **Surface to surface distance (mm) visualization**: *Shape reconstruction error interpolated as a heatmap on ground truth reconstructed meshes. Best, Median and Worst case result on LocalizedSSM.*

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

## Appendix A. Architecture Details

### A.1. Segmentation

A 3D UNet is used for segmentation. It consists of five downsampling, five upsampling layers, and a bottleneck in between. In downsampling/encoder layers, each layer consists of two convolution layers with kernel size 3x3 and padding 1 followed by a ReLU activation and 3D Batch Norm and then a 3D Max Pool of kernel size 2. We start with the initial feature output of the first layer as 16 and double it every layer till the end of the downsampling layer. For the decoder / upsampling layers, each layer has a 3D convolution transpose operation that preserves the features and acts as an inverse to max pool 3D; it uses a kernel of size two and stride two followed by two convolution layers with kernel size 3x3 and padding 1, followed by a ReLU activation and 3D Batch Norm, same as in encoder. The only difference is that the size of channels is reduced to divide by 2 for every layer, and it also takes the corresponding input from the encoder (skip connection) for each layer. Finally, the last decoder layer has two convolution layers with kernel size 3x3 that outputs the required channels (both input and output channels are 1), followed by a sigmoid.

### A.2. Spatial Transformer

We concatenate the cropped anatomy and the mean segmentation channel-wise to pass it to shape registration subnetwork to estimate the 12-parameter theta matrix (3 for translation and 3x3 rotation). The shape registration subnetwork, also referred to as a localization net in the context of spatial transformers, uses the same architecture as UNet Encoder with 5 layers and the starting feature output of the first layer as 8. We then pass the output from the encoder to another linear layer to give 12 values. For this final layer, we initialize the parameters corresponding to translation as 0 and the parameters of the rotation matrix as identity. We then create an affine grid of the same dimensions as the cropped region using

the theta matrix; the grid is generated using the function provided by PyTorch. We use the grid to sample the cropped anatomy, which transforms it based on the theta values

### A.3. Shape Subnetwork

The shape subnetwork architecture is the same as DeepSSM. CNN layers with five convolution layers followed by two fully connected layers to produce the output regression coefficients. The input to the network is the cropped 3D image combined with a Gaussian filter (the Gaussian filter provides an attention map for the shape regression subnetwork), and the output is a set of ordered PCA loadings with respect to the shape space. The PCA loadings are then passed to a fixed linear decoder, whose weights are fixed to the Eigen-matrix and bias to mean shape, to give the world particles.

## Appendix B. Cropping

The segmentation output is essentially a binary 3D image where pixels not corresponding to the anatomy of interest are zeros. We use this mask to get the bounding box of the region with non-zeros pixel values and use it to crop the segmentation and, correspondingly, the input image. Then finally, resize them to a constant size of $64 \times 64 \times 64$.

## Appendix C. Training Details

The batch size is set to 1, $\lambda_{seg} = 100$ is fixed. $\lambda_{reg}$ is set to 100 after ten epochs. And after ten more epochs, we keep on increasing $\lambda_w$ to a maximum value of 10 over the duration of 10 epochs. We have determined these values after extensive experimentation and cross-validation. Also, note that the segmentation network is kept completely separated; we break the gradient flow after segmentation so $L_{reg}$ (registration network) and $L_w$ (shape subnetwork) does not affect the segmentations (same as segmentation trained independently). We do this because we found out in our experiments that propagating the $L_{reg}$ hurts the segmentation performance. A step learning rate scheduler is used with gamma 0.99 and updated every 50 epochs.

