# OpenReview forum: "Localization-aware Deep Learning Framework for Statistical Shape Modeling Directly from Images"
_MIDL.io/2023/Conference — MIDL 2023 Poster_

### Official Review · Reviewer_HtHP · 2023-02-02

**Confidence:** 2
**Preliminary Rating:** 2

**Summary:**

This paper presents a dl-framework for statistical shape modeling which learns simultaneously to segment the anatomy of interest, estimate a rigid transformation to the mean shape and predict the corresponding statistical representations. The evaluation is performed on two datasets: the synthetic Supershape dataset and a left atrium MRI dataset (that is not further specified). The results indicate that the method works slightly better than the comparison method (on which this proposed method is built on).

**Strengths:**

- The paper is well-written and mostly easy to follow.
- The results indicate that the proposed method performs slightly better than the comparison method.
- The authors show the best, average, and worse performing cases in a figure to give an impression of the results.


**Weaknesses:**

- The discussion including the limitations of the presented method is completely missing.
- The scientific value is limited. It seems that the “only” added value is the combination of all components into one training procedure.
- If I understood correctly, the “ground truth” for the PDM model has also been generated automatically. It is not clear to me why we need this need tool then. And how much does the evaluation depends on the actual accuracy of the reference method? How good is the reference method?
- It is not clear what the differences of the shape sub-network are to the work of Bhalodia. “We based our shape sub-network on DeepSSM (Bhalodia et al., 2018a).”


**Deanonymize Review:**

no

**Paper Type:**

methodological development

**Questions To Address In The Rebuttal:**

- Add a discussion about the limitations and advantages of the presented method. How does it differ from DeepSSM and why does it perform better?
- Is there any public dataset that is typically used to compare SSM methods against each other? If so, please run your model on that dataset.

---

### Official Review · Reviewer_vvEf · 2023-02-03

**Confidence:** 4
**Preliminary Rating:** 4
**Recommendation:** Poster

**Summary:**

This paper presents an end-to-end deep learning framework for statistical shape modeling directly from images, consisting of segmentation, registration, and shape regression networks. Based on the baseline DeepSSM, this work further alleviates manual input burdens and automates the pipeline of statistical shape modeling.

Experiments on two datasets show that the qualitative and quantitative performance of this method is comparable to the baselines.

**Strengths:**

+ As a multi-task network, the deep framework Image2SSM, could simultaneously segment the anatomy of interest, perform the rigid transformation to align to the mean shape, and predict the statistical representation of the anatomy.
+ The experimental verifications are reasonable.
+ The paper is well-written and easy to follow.

In general, the ideas and practices of this paper are relatively fresh in the field. However, I still have some concerns regarding this paper.

**Weaknesses:**

+ Experimentations show that the qualitative and quantitative improvement of the proposed Image2ssm compared with baselines is not significant, especially in Fig2(a) and Fig4(b).

+ In my opinion, the DeepSSMc using cropped images can be viewed as upper bound with ground truth segmentation, and the DeepSSMf using full images be viewed as lower bound without segmentation. On the Left Atrium dataset, Fig4 shows that the DeepSSMf has a better performance. Please explain this counterintuitive phenomenon.

+ The author should verify the effectiveness and necessity of the registration module, and also demonstrate localization awareness.

**Deanonymize Review:**

no

**Detailed Comments:**

+ Fig.1 needs to be improved. To make it clear, its font size should be adjusted.
+ Each formula should end with a period.

**Paper Type:**

both

**Questions To Address In The Rebuttal:**

Please address the concerns mentioned in the Weaknesses part.

+ Experimentations show that the qualitative and quantitative improvement of the proposed Image2ssm compared with baselines is not significant, especially in Fig2(a) and Fig4(b).

+ In my opinion, the DeepSSMc using cropped images can be viewed as upper bound with ground truth segmentation, and the DeepSSMf using full images be viewed as lower bound without segmentation. On the Left Atrium dataset, Fig4 shows that the DeepSSMf has a better performance. Please explain this counterintuitive phenomenon.

+ The author should verify the effectiveness and necessity of the registration module, and also demonstrate localization awareness.

---

### Official Review · Reviewer_9QZ9 · 2023-02-04

**Confidence:** 4
**Preliminary Rating:** 2
**Recommendation:** Poster

**Summary:**

The authors present a deep-learning-based pipeline to automatically detect and segment anatomies of interest, which is trained end-to-end. Specifically, the pipeline comprises three subnetworks:
(1) a UNet for object detection and segmentation,
(2) a spatial transformer network to transform the object into the canonical shape space, and
(3) DeepSSM-based regression of low-dimensional shape descriptors and point correspondences.

The method, evaluated on a synthetic and a real-world dataset, shows similar or better performance compared to directly applying DeepSSM on the 3D input volumes.

**Strengths:**

- The paper provides a good description of the research gap and is easy to read.

- The method is particularly interesting since the presented method can directly infer the anatomy segmentation, i.e., without manual interactions, e.g., cropping or drawing a tight bounding box around the object of interest.

- To me, the most important and interesting component of the presented method is: (1) the explicit constraint of the segmentation by estimating the shape coefficients of a statistical shape model, and (2) the integration of a registration subnetwork, which brings the detected object into the canonical pose of the statistical shape model and thus improves the estimation of shape coefficients.

**Weaknesses:**

- The paper lacks a proper discussion of the results
- In its current form, the paper lacks explanations and other crucial information to reproduce the results
- Additional quantiative results and/or experiments would be necessary to fully appreciate and understand the impact of each component
- There are some errors in both text and figures
- The presentation of figures and shape visualizations could be improved
- The novelty of the proposed method is somewhat limited


**Deanonymize Review:**

no

**Detailed Comments:**

In general, the presented paper is interesting to read, but there are several issues that should be addressed:

The paper lacks a proper discussion of the results as well as of the limitations of the current approach. I would kindly ask the authors to make space for it, e.g., by showing less examples in Fig. 3 & 5 or by moving them to the supplementary material.

While I strongly believe that the proposed method has several merits, it is not clear to me to what extent the presented pipeline is novel:
1. the whole pipeline comprises well-established methods with presumably little adaptation.
2. robust initialization of statistical shape models has been actively researched over the last decades. One particularly relevant method that shares several similarities with the presented method is presented by Zheng et al. (2007), “Fast automatic heart chamber segmentation from 3D CT data using marginal space learning and steerable features”, with the deep-learning extension presented by Ghesu et al. (2016). This method addresses finding the position, rotation, and scale information of anatomies, and is even able to fit a statistical shape model to the images.

Moreover, the method section could benefit from additional explanations and information since the current description is often incomplete and does not allow reproducibility of results.
- Segmentation subnetwork: For reproducibility, please provide a detailed description of the UNet architecture, i.e., number of layers, number of features, activation functions, choice of down-/upsampling operation, etc.

- Registration subnetwork: Since the paper mentions both rigid and affine registration, I would kindly ask the authors to clarify which registration was used. In addition, I would advice the authors to add a short, but complete, description of how the spatial transformer works. At least, I would ask for: one equation showing the formula of the transformation and the free parameters; a description of the network architecture; a brief explanation of the grid generator that is illustrated in Fig. 1 (but that is never mentioned in text)

- Page 3, Section 2.2: Please clarify whether \bar{S} denotes the mean shape of the PDM or the mean over all segmentation masks. For the latter, please clarify whether the result is a real-valued mask or a binarized version of it – please then also justify the choice of using it as the target within the supervised training. Similarly, a comment on choosing the MSE loss over other losses, e.g., the Dice loss, would be appreciated.

- Shape subnetwork: The description of the network architecture (2 + 1 fully connected layers) does not match the graphic and neither the referenced DeepSSM. Please revise the text and add a full description of the network architecture.

- Page 4, Section 2.4: Please provide additional information about the network training for reproducibility of results, i.e., how and to what maximum value are \lambda_reg and \lambda_w increased, batch size, learning rate decay factor, etc.

- Page 4, Section 3: How is the surface-to-surface distance defined? Is it the (average) Haussdorff distance?

In addition, there are several experiments that would greatly strengthen the paper:

- It would be great to see whether end-to-end training provides significant improvement over training each subnetwork separately or successively. At least, however, please provide some quantitative results on the segmentation & registration performance, to put the final reconstruction results into perspective. It may also be worth to show the performance of the initial PDM on the test set.

- I would be particularly interested to see what the performance difference is between the presented pipeline and a modified version where the shape regression network is trained on matching the cropped & transformed segmentation mask to the shape descriptors. Could the authors please share their thoughts on this?


Minor comments:
- Fig. 1: Labels S’’ and I’’ are swapped
- Page 4, Section 2.3: The text does not match Eq. 3 and/or Eq. 3 is wrong (e.g. Gaussian kernel in 3D should have a weight of 1 / (sqrt(2 \pi)^3 \sigma^3). One note: It may be much easier to express the filter operation as a convolution). Please also clarify whether the segmentation mask is a binary mask or a probability mask and justify why filtering with a Gaussian kernel is necessary to obtain an attention map.
- Fig. 1: It is not clear on first sight whether “PCA” within the shape module denotes applying PCA or the shape coefficients. Please revise
- P.2, Section 1 & 2: The description of the algorithm pipeline is repetitive. I would suggest to keep the description of the pipeline within the introduction concise and then give a high-level overview of the pipeline at the beginning of Section 2.
Please also emphasize that this work requires / relies on a pre-build PDM.
- P.2, Section 2: Please define H, W, D
- P.3, Section 2.1: Please clarify how the cropping is done (I assume by defining the axis-aligned bounding box)
- P.3, Eq. 1: L_seg is not integrated into the text
- P.5, Section 3.1: Please add a reference to the supershape generation algorithm or show the mathematical formula. Please comment on how large the supershapes are in relation to the volume size. Please also mention the amplitude of the additive Gaussian noise
- Fig. 2 & 4: Y-axis labels would improve readability of the figure
- Fig. 3 & 5: It may be beneficial to show or overlay the ground truth shape, to visually inspect the differences
- P. 5, Section 3.2: Please add a reference to the Left Atrium MRI dataset

*Nevertheless, I believe that this work, in a revised version however, would be of great interest to the community.*


**Paper Type:**

both

**Questions To Address In The Rebuttal:**

Please address the problems and concerns under “Weaknesses” and “Detailed comments”.

In particular, I would kindly ask the authors to move Fig. 3 (and possibly also Fig. 5) to the supplementary material and use the free space to strengthen the method section and add a discussion of the results as well as the current limitations.

---

### Meta-Review · Area_Chair_fwTP · 2023-02-23

**Recommendation:** Accept (Poster)
**Confidence:** 2

**Metareview:**

In this paper the authors propose an end to end tool for shape modelling that automatically crops, segments, registered and extracts a shape modelling. Reviewers criticise the paper for limited novelty but did not respond to the authors detailed response to their critiques. I am on the fence. I agree with the authors that they are offering novelty through putting a pipeline together that automates what can be a time costly and error prone process. The results seem to show that they are not losing any accuracy doing that, which is good. I think its just a shame they couldn’t more clearly articulate the benefits -  perhaps performance on edge cases or on run time. In the end, one might argue that the difference with the DeepSMM does seem to reduce just to an affine pre alignment that should not be that time costly or difficult a preprocessing step. In summary, I think this is a clearly written paper, well motivated, the method works well, but perhaps the results could have been more convincing. It could perhaps be an interesting poster if the authors could better emphasise its strengths.